# Explicit Stable Finite Difference Methods for Diffusion-Reaction Type Equations

Humam Kareem Jalghaf [1,2], Endre Kovács [3,*], János Majár [3], Ádám Nagy [3] and Ali Habeeb Askar [1,2]

1   Department of Fluid and Heat Engineering, University of Miskolc, 3515 Miskolc, Hungary; 20310@uotechnology.edu.iq (H.K.J.); 20156@uotechnology.edu.iq (A.H.A.)
2   Mechanical Engineering Department, University of Technology, Baghdad 10066, Iraq
3   Institute of Physics and Electrical Engineering, University of Miskolc, 3515 Miskolc, Hungary; fizmajar@uni-miskolc.hu (J.M.); fizadam@uni-miskolc.hu (Á.N.)
*   Correspondence: fizendre@uni-miskolc.hu or kendre01@gmail.com

**Abstract:** By the iteration of the theta-formula and treating the neighbors explicitly such as the unconditionally positive finite difference (UPFD) methods, we construct a new 2-stage explicit algorithm to solve partial differential equations containing a diffusion term and two reaction terms. One of the reaction terms is linear, which may describe heat convection, the other one is proportional to the fourth power of the variable, which can represent radiation. We analytically prove, for the linear case, that the order of accuracy of the method is two, and that it is unconditionally stable. We verify the method by reproducing an analytical solution with high accuracy. Then large systems with random parameters and discontinuous initial conditions are used to demonstrate that the new method is competitive against several other solvers, even if the nonlinear term is extremely large. Finally, we show that the new method can be adapted to the advection–diffusion-reaction term as well.

**Keywords:** UPFD method; diffusion equation; heat transfer; explicit time-integration; stiff equations; unconditional stability

## 1. Introduction

We are going to study the following diffusion-reaction equation:

$$\frac{\partial u}{\partial t} = \alpha \nabla^2 u - Ku + q - \sigma u^4, \tag{1}$$

where $u$ is the unknown concentration, while the known parameters are the diffusion coefficient $\alpha$, the coefficient of the linear reaction term $K$, and the source of particles $q$ in case of particle diffusion [1]. It is well known that this equation can describe heat transfer as well. In this case, the physical meaning of $u$ is the temperature and $\alpha$ is the thermal diffusivity. If one uses Newton's law of cooling, then heat transfer by convection can be expressed [2] (Equation (3)) by a term proportional to $u_a - u$, where $u_a$ is the ambient temperature which can be considered as constant. It means that $-Ku + q$ can represent conductive heat transfer. Moreover, heat generation by chemical reactions, radioactive decay, etc. can also be incorporated into $q$. According to the Stefan–Boltzmann law [3] (Chapter 8), the heat transfer due to radiation can be taken into account by a term proportional to $u_a^4 - u^4$, where the proportionality constant includes the surface area and the Stefan–Boltzmann constant, all of which are nonnegative quantities. The $u_a^4$ term can obviously be included into the heat source term $q$. Therefore, in this paper, we will use the phrase *radiation term* when we talk about the fourth order term $-\sigma u^4$.

At the end of the paper, we will also examine the linear advection–diffusion-reaction equation in its most standard form:

$$\frac{\partial u}{\partial t} = \alpha \frac{\partial^2 u}{\partial x^2} u - a \frac{\partial u}{\partial x} - Ku. \tag{2}$$

If the medium is not homogeneous, more general forms of the above equations must be used. For example, Equation (1) can be generalized as

$$\frac{\partial u}{\partial t} = \frac{1}{c\rho}\nabla(k\nabla u) - Ku + q - \sigma u^4,\tag{3}$$

where, in case of conductive heat transfer, $k = k\left(\vec{r},t\right) \geq 0$, $c = c\left(\vec{r},t\right) > 0$ and $\rho = \rho\left(\vec{r},t\right) > 0$ are the heat conductivity, specific heat and mass density, respectively. The $\alpha = k/(c\rho)$ relation holds between these quantities. Besides, $K = K\left(\vec{r},t\right) \geq 0$ and $q = q\left(\vec{r},t\right)$ are also known functions of the space and time coordinates.

There are plenty of numerical methods to solve these equations, such as several versions of the finite difference methods (FDM) [4,5], finite element methods (FEM) [6] or a combination of these [7]. However, these require the full spatial discretization of the system, thus they can be computationally demanding. Scientists and engineers must also deal with systems where the physical properties such as the heat conductivity can be highly different [8] (p. 15), even in regions of the system which are in the vicinity of each other. This implies that the coefficients such as $\alpha$ or $k$, and hence the eigenvalues of the problem may have a range of several orders of magnitude, which means that the problem can be rather stiff.

We explained in our previous papers [9–11], and also will demonstrate in this paper that the widely used conventional solvers, either the explicit or the implicit ones, have serious difficulties when they are used for these problems. The explicit methods are usually only conditionally stable, so very small time step sizes have to be used [12], especially if the stiffness of the problem is high. This is the main reason that these equations are typically solved by implicit methods, as it has been conducted by Zhang and Zhao [13], Aamoah-Mensah et al. [14], Nana and Munyakazi [15], Aminikhah and Alavi [16], Ali et al. [17] and Singh et al. [18]. The authors of [14] claim that the larger computational cost of implicit methods is compensated by their unrestricted stability. We think that this is perfectly true when the number of cells is small, such as in their study, but when this number is very large, which is almost always the case in two and especially in three space-dimensions, the implicit solvers become extremely slow with huge memory usage.

Nevertheless, the search for and the application of alternative methods are continuous. The most important example are the nonstandard finite difference schemes (NSFD) introduced by Mickens [19], which have been applied by others to different equations containing the diffusion term [20–22]. Since the formerly fast increase of the CPU clock frequencies halted more than a decade and a half ago, and the trend towards increasing parallelism in high performance computing is reinforced [23,24], we believe that the easily parallelizable explicit and unconditionally stable methods for numerically solving these equations will have an increasing role in the future. Albeit these methods are less known, some scholars work with them. For example, Karahan used explicit Sauljev-type alternating direction explicit (ADE) method for the advection–diffusion equation [25]. Sanjaya and Mungkasi examined the performance of the same method and found that it is indeed accurate [26]. Pourghanbar et al. used ADE/Sauljev to solve a fully nonlinear PDE [27]. Harley applied the odd-even hopscotch method to the Frank-Kamenetskii reaction-diffusion equation [28]. Then Al-Bayati et al. compared the ADE, the alternating direction implicit (ADI) and the Hopscotch method in the case of the Gray–Scott reaction-diffusion equation [29]. These methods perform quite well in an equidistant and regular mesh, but they heavily rely on these beneficial properties of the mesh.

One of the very few alternatives which can be applied for a general mesh is the unconditionally positive finite difference (UPFD) scheme of Chen-Carpentier and Kojouharov [30]. It is reported that this method is indeed completely stable even for stiff systems [31], but its accuracy is far from being optimal [32], even compared to other first order methods, such as the standard explicit (Euler) FDM method [32,33] or our recently invented constant-neighbor method [31]. In this paper, we construct and test a similar

method than the UPFD, but it consists of two stages, and therefore it is significantly more accurate. Our current work was inspired by the so-called theta formulas, where the user can adjust the extent of implicitness by the parameter $\theta$.

The paper is structured as follows. In Section 2, we first introduce the new algorithm for the simplest, one dimensional, equidistant mesh. Then, in Section 2.2, the analytical investigation of the convergence and stability properties of this method is explained. After this, we present the generalization of the algorithm for arbitrary meshes. In Section 3, we begin with the verification of the new method by comparing it to an analytical solution and the heat conduction equation with the radiation term. Then, in Sections 3.2 and 3.3, two numerical experiments are presented for two space dimensional stiff systems consisting of 12,000 cells without and with the nonlinear radiation term, respectively. In Section 3.4 we make attempts to apply the new method to the advection–diffusion-reaction equation. In Section 4, we summarize the conclusions and sketch our future research plans.

## 2. The New Method

### 2.1. Construction of the New Method

In one space dimension, we take $x_i = i\Delta x$, $i = 0, \ldots, N-1$, which is a common space discretization. Let us fix the time discretization to $t_n = t_0 + nh$, $n = 0, \ldots, T$, $T = (t_{\text{fin}} - t_0)/h$. We introduce the mesh-parameter $r = \frac{\alpha h}{\Delta x^2}$ and $\mu = \frac{ah}{\Delta x}$. The original UPFD method applies the most common [34] (p. 112) spatial discretization of the diffusion term based on the central difference formula, while it applies the backward difference formula for the advection term. However, the time levels are treated in a tricky way [32], such that the neighbors are taken into account fully at the old time level, where their values are known, and only the actual cell is treated implicitly. It means that for example $u_{i-1}^n$ is used instead of $u_{i-1}^{n+1}$, with which they obtained:

$$\frac{u_i^{n+1} - u_i^n}{h} = \alpha \frac{u_{i-1}^n - 2u_i^{n+1} + u_{i+1}^n}{\Delta x^2} - a \frac{u_i^{n+1} - u_{i-1}^n}{\Delta x} - K u_i^{n+1}, \tag{4}$$

This can be arranged to a fully explicit form to obtain the following Algorithm 1:

---
**Algorithm 1:** The original UPFD

$$u_i^{n+1} = \frac{u_i^n + r\left(u_{i-1}^n + u_{i+1}^n\right) + \mu u_{i-1}^n}{1 + 2r + \mu + Kh} \tag{5}$$

---

Now we adapt this method to Equation (1) where $a = 0$ but $\sigma > 0$. In principle the nonlinear term can be incorporated into this scheme in many different ways. We choose the following treatment: We insert the radiation term at the level of Equation (4) as $u_i^4(t) \approx u_i^{n+1}\left(u_i^n\right)^3$, which again can be expressed in an explicit form, and with this we obtain the following adaptation of the original UPFD algorithm to Equation (3) (Algorithm 2):

---
**Algorithm 2:** UPDF for the diffusion-reaction-radiation Equation (3)

$$u_i^{n+1} = \frac{u_i^n + r\left(u_{i-1}^n + u_{i+1}^n\right) + q_i h}{1 + 2r + K_i h + \sigma h \left(u_i^n\right)^3}$$

---

If $r$, $q_i$, $K_i$ and $\sigma$ have arbitrary nonnegative values and the values of $u$ at the beginning of the time stapes are nonnegative, then both the numerator and the denominator is nonnegative in this formula. It means that this formula preserves positivity similarly to the original UPFD formula for the strongly nonlinear case as well. As we will see later, its accuracy is not very good, thus we proceed to construct a two-stage method as well.

We are going to combine the UPFD idea with the so called $\theta$-method, which can be applied for the diffusion term in the following way:

$$u_i^{n+1} = u_i^n + r\left[\theta\left(u_{i-1}^n - 2u_i^n + u_{i+1}^n\right) + (1-\theta)\left(u_{i-1}^{n+1} - 2u_i^{n+1} + u_{i+1}^{n+1}\right)\right], \qquad (6)$$

where $\theta \in [0,1]$. If $\theta = 1$, this scheme is the forward-time central-space (FTCS) scheme, which is basically the explicit Euler time integration. For smaller values of $\theta$, this formula is implicit, and for $\theta = 0, \frac{1}{2}$ one has the implicit (Euler) and the Crank–Nicolson method, respectively [35]. Using the trick above and incorporating the reaction and the source terms we can write:

$$u_i^{n+1} = u_i^n + r\left[-2\theta u_i^n - 2(1-\theta)u_i^{n+1} + u_{i-1}^n + u_{i+1}^n\right] - hK_i u_i^{n+1} + hq_i + \sigma u_i^{n+1}(u_i^n)^3. \quad (7)$$

If one takes $\theta = 0$, the original UPFD treatment is obtained back. The point is that this more general formula can also be easily rearranged to obtain an explicit formula, according to which the new value of the $u$ variable has the following form in the 1D equidistant case (Algorithm 3):

---

**Algorithm 3:** Theta-generalization of Algorithm 2

$$u_i^{n+1} = \frac{(1-2r\theta)u_i^n + r\left(u_{i-1}^n + u_{i+1}^n\right) + hq_i}{1 + 2r(1-\theta) + hK_i + \sigma h(u_i^n)^3} \qquad (8)$$

---

Since we formally started from an implicit Formula (6) but made it fully explicit, we started to call these methods *pseudo-implicit*. The main novelty of this paper is that we organize Formula (8) into a two-stage method as follows inspired by the well-known predictor-corrector methods [35–37]. The calculation starts with taking a fractional-sized time step using the already known $u_i^n$ values, and then a full time step is made Algorithm 4.

---

**Algorithm 4:** 2-stage pseudo-implicit method for the diffusion-reaction-radiation Equation (1)

*Stage 1.* Take a partial time step $h_1 = ph$, $p > 0$ using Formula (8) with parameter $\theta_1$:

$$u_i^{\text{pred}} = \frac{(1-2pr\theta_1)u_i^n + pr\left(u_{i-1}^n + u_{i+1}^n\right) + q_i h_1 - v_1 K_i h_1 u_i^n}{1 + 2pr(1-\theta_1) + v_2 K_i h_1 + \sigma h_1 (u_i^n)^3}$$

*Stage 2.* We redefine $u_i^{\text{pred}}$ by calculating the linear combination with $0 < \lambda \le 1$:

$$u_i^{\text{pred}} = \lambda u_i^{\text{pred}} + (1-\lambda)u_i^n \qquad (9)$$

---

Take a full time step with the (8) formula with parameter $\theta_2$:

$$u_i^{n+1} = \frac{(1-2r\theta_2)u_i^n + r\left(u_{i-1}^{\text{pred}} + u_{i+1}^{\text{pred}}\right) + q_i h - K_i h\left(w_1 u_i^n + w_2 u_i^{\text{pred}}\right)}{1 + 2r(1-\theta_2) + (1-w_1-w_2)K_i h + \sigma h\left(u_i^{\text{pred}}\right)^2 u_i^n},$$

where $v_1$, $v_2$, $w_1$, $w_2$ are real numbers which are considered as free parameters. We must mention that the mathematically correct form of (9) would be $u_i^{\text{lin}} = \lambda u_i^{\text{pred}} + (1-\lambda)u_i^n$, but we immediately put down it in the form which is to be used in a computer code to spare memory. We also note that with this treatment of the nonlinear term we obtain a second-order method with very good stability properties, as we will see later.

### 2.2. Analytical Investigations

We perform the calculations of Algorithm 4 for the linear $\sigma = 0$ case. First, we express the new values $u_i^{n+1}$ by the old values $u_j^n$. For this we calculate the linear combination (9) with the Mathematica software. If we use notations $\kappa_i = K_i h$ and $\psi = 1 - \theta_1$, then we have

$$
\begin{aligned}
u_i^{\text{pred}} &= \lambda \frac{(1-2pr\theta_1)u_i^n + pr\left(u_{i-1}^n + u_{i+1}^n\right) + q_i ph - v_1 p\kappa_i u_i^n}{1 + 2pr\psi + v_2 p\kappa_i} + (1-\lambda)u_i^n \\
&= \left[\frac{\lambda(1-2pr\theta_1 - v_1 p\kappa_i)}{1 + 2pr\psi + v_2 p\kappa_i} + (1-\lambda)\right]u_i^n + \frac{\lambda p}{1 + 2pr\psi + v_2 p\kappa_i}\left(ru_{i+1}^n + ru_{i-1}^n + hq_i\right),
\end{aligned}
$$

which yields

$$
\begin{aligned}
u_i^{n+1} =\ & \frac{1}{[1+2pr\psi+v_2 p\kappa_i][1+2r(1-\theta_2)+(1-w_1-w_2)\kappa_i]}\Big\{\lambda pr^2\, u_{i-2}^n \\
&+ r[1 + 2pr(\psi-\lambda) + p\kappa_i[v_2 - \lambda(v_1 + v_2 + w_2)]]u_{i-1}^n \\
&+ [1 - 2r\theta_2 + 2pr[-2(r\theta_2 - 1)\psi - \psi + r\lambda] + \kappa_i[(1+2pr\psi)(w_1+w_2) - p(v_2 + 2r(w_2\lambda - v_2\theta_2))] \\
&- p\kappa_i^2[\lambda w_2(v_1+v_2) - v_2(w_1+w_2)]]u_i^n + r[1 + 2pr(\psi-\lambda) + p\kappa_i[v_2 - \lambda(v_1+v_2+w_2)]]u_{i+1}^n \\
&+ \lambda pr^2 u_{i-2}^n + \lambda hprq_{i-1} + h[1 + 2pr\psi + p\kappa_i(v_2 - w_2\lambda)]q_i + \lambda hprq_{i+1}\Big\}.
\end{aligned} \tag{10}
$$

After spatial discretization as discussed above, Equation (1) for $\sigma = 0$ can be written into a brief matrix-form:

$$
\frac{d\vec{u}}{dt} = M\vec{u} + \vec{q}. \tag{11}
$$

The system matrix $M$ is tridiagonal (in the one-dimensional case) and it is the sum of two terms related to the diffusion and the linear reaction terms, respectively:

$$
M = M^D + M^R. \tag{12}
$$

In the equidistant case, these matrices have the following elements:

$$
M_{ii}^D = -\frac{2\alpha}{\Delta x^2}\ ,\quad M_{i,i+1}^D = \frac{\alpha}{\Delta x^2}\ ,\quad M_{i,i-1}^D = \frac{\alpha}{\Delta x^2}\ ,\quad M_{ii}^R = -K_i. \tag{13}
$$

Let us start with the analysis of the convergence properties of the methods in the one-dimensional case for constant values of the mesh ratio $r$, in other words, when the equidistant spatial discretization is fixed.

**Theorem 1.** *For $\sigma = 0$, the order of convergence of the Algorithm 4 is two for the system (6) of linear ODEs:*

$$
\frac{d\vec{u}}{dt} = M\vec{u} + \vec{q}\ ,\quad \vec{u}(t=0) = \vec{u}^0, \tag{14}
$$

*where M is defined in (11)–(13), if and only if the conditions*

$$
p\lambda = \frac{1}{2}, \tag{15}
$$

$$
\theta_2 = \frac{1}{2}, \tag{16}
$$

$$
2w_1 + w_2 = 1, \tag{17}
$$

$$
v_1 + v_2 = 1, \tag{18}
$$

*hold, where M is introduced in (11)–(13), while $\vec{q}$ and $\vec{u}^0$ are arbitrary vectors.*

**Proof.** We will show that the local error of the numerical solution (10) compared to the analytical solution

$$\vec{u}^{n+1} = e^{Mh}\vec{u}^n + \left(e^{Mh} - 1\right)M^{-1}\vec{Q} = \left(1 + Mh + M^2\frac{h^2}{2} + M^3\frac{h^3}{3!} + \dots\right)\vec{u}^n + \left(h + M\frac{h^2}{2} + M^2\frac{h^3}{3!} + \dots\right)\vec{Q}.$$

of the ODE system (14) is less than second order in the time step size $h$. Let us introduce the notation $\beta = r/h$ and express the exact solution in series up to second order in $h$:

$$u_i^{n+1} = 1 + \left\{\beta u_{i-1}^n - [K_i + 2\beta]u_i^n + \beta u_{i+1}^n + q_i\right\}h$$
$$+ \left\{\frac{\beta^2}{2}u_{i-2}^n - \left[2\beta^2 + \frac{K_i\beta}{2} + \frac{K_{i-1}\beta}{2}\right]u_{i-1}^n + \left[3\beta^2 + 2\beta K_i + \frac{K_i^2}{2}\right]u_i^n - \left[2\beta^2 + \frac{K_i\beta}{2} + \frac{K_{i+1}\beta}{2}\right]u_{i+1}^n\right.$$
$$\left. + \frac{\beta^2}{2}u_{i+2}^n + \frac{\beta}{2}q_{i-1} - \left[\beta + \frac{K_i}{2}\right]q_i + \frac{\beta}{2}q_{i+1}\right\}h^2 + O(h^3).$$

If we do the same with the numerical solution (10), we obtain:

$$u_i^{n+1} = 1 + \left\{\beta u_{i-1}^n - [K_i + 2\beta]u_i^n + \beta u_{i+1}^n + q_i\right\}h$$
$$+ \left\{\left[p\beta^2\lambda\right]u_{i-2}^n - \left[2\beta^2(1 - \theta_2 + p\lambda) + K_i\beta(pw_2\lambda + 1 - w_1 - w_2) + K_{i-1}p\beta\lambda(v_1 + v_2)\right]u_{i-1}^n\right.$$
$$+ \left[2\beta^2(2 - 2\theta_2 + p\lambda) + 2\beta K_i[2 - \theta_2 - w_1 - w_2(1 - p\lambda)] + K_i^2[(1 - w_1 - w_2) + pw_2\lambda(v_1 + v_2)]\right]u_i^n \qquad (19)$$
$$- \left[2\beta^2(1 - \theta_2 + p\lambda) + K_i\beta(pw_2\lambda + 1 - w_1 - w_2) + K_{i+1}p\beta\lambda(v_1 + v_2)\right]u_{i+1}^n + \left[p\beta^2\lambda\right]u_{i+2}^n$$
$$\left. + [p\beta\lambda]q_{i-1} - [2\beta(1 - \theta_2) + K_i(pw_2\lambda + 1 - w_1 - w_2)]q_i + [p\beta\lambda]q_{i+1}\right\}h^2 + O(h^3).$$

It can be immediately seen that the two expressions coincide up to first order, but some conditions must be fulfilled for second order equality. From the equality requirement of the coefficients of $u_{i\pm2}^n$, and $q_{i\pm1}$, we have $p\beta^2\lambda = \frac{\beta^2}{2}$ and $p\beta\lambda = \frac{\beta}{2}$, thus (15) must hold. The same requirement for $q_i$ gives

$$2\beta(1 - \theta_2) + K_i(pw_2\lambda + 1 - w_1 - w_2) = \beta + \frac{K_i}{2},$$

which yields (16) and (17) if one uses condition (15). Now we substitute back the conditions obtained until now to the coefficients of $u_{i\pm1}^n$ and $u_i^n$, and obtain

$$-\left[2\beta^2 + \frac{K_i\beta}{2} + \frac{K_{i\pm1}\beta}{2}(v_1 + v_2)\right] = -\left[2\beta^2 + \frac{K_i\beta}{2} + \frac{K_{i\pm1}\beta}{2}\right]$$
$$3\beta^2 + 2K_i\beta + \frac{K_i^2}{2}[2w_1 + (1 - 2w_1)(v_1 + v_2)] = 3\beta^2 + 2\beta K_i + \frac{K_i^2}{2},$$

which yields (18), and the proof is now complete. □

We note that our original idea was to organize the original UPFD formula into a two-stage method, which means $\theta_1 = 0$, $\theta_2 = 0$. With that idea we obtained a new unconditionally positive method, but it was only first and not second order. After Theorem 1 and its proof one can understand the reason of this failure.

Before starting to analyze the stability of the methods, we also note that in the case of the original UPFD Algorithm 1 for $K = 0$, the new $u_i^{n+1}$ values are the convex combinations of the old $u_j^n$ values which immediately implies not only unconditional stability, but the positivity preserving property, too. If $K$ is increased to any positive number, this latter property still holds since $K$ is in the denominator with a positive sign. However, the new Algorithms 3 and 4 contain the theta-formula for $\theta \neq 0$, thus they cannot be not positivity-preserving. This is the price we have to pay for second order accuracy.

We are going to use the most standard von Neumann stability analysis (see [38], Chapter 8, as well as [39]) to prove the unconditional stability of the method in the linear case. For this, the $u_i^n$ values in the expression of $u_i^{n+1}$ must be replaced by the errors $\varepsilon_i^n$ and the error-function is decomposed into a Fourier series as follows:

$$\varepsilon_i^n = \sum_m E_m(t)e^{Ik_m x}, \quad \varepsilon_{i\pm1}^n = \sum_m E_m(t)e^{Ik_m(x\pm\Delta x)},$$

where $E_m(t)$ is the amplitude of the $m$-th term $e^{Ik_m x}$ in the Fourier series of the error and $I$ is the imaginary unit $\sqrt{-1}$. We omit the $m$ index for the sake of simplicity, and introduce the notation $\gamma = k_m \Delta x$, with which we obtain the following relations:

$$\varepsilon_i^{n+1} = E(t+h)e^{Ikx} \text{ and } \frac{\varepsilon_{i-1}^n + \varepsilon_{i+1}^n}{2} = \frac{E(t)e^{Ikx}\left(e^{-I\gamma} + e^{I\gamma}\right)}{2} = E(t)e^{Ikx}\cos\gamma,$$

$$\frac{\varepsilon_{i-2}^n + 2\varepsilon_i^n + \varepsilon_{i+2}^n}{2} = 2E(t)e^{Ikx}\cos^2\gamma,$$

Using these expressions and simplifying with $e^{Ik_m x}$ one can obtain the amplification factor, which is defined as $G = \frac{E(t+h)}{E(t)}$. If this factor is in the closed interval $[-1, 1]$ for arbitrary time step size $h$, then the errors cannot be amplified regardless of how large $h$ one uses, which means unconditional stability.

**Theorem 2.** *If $K = 0$ and $\sigma = 0$, then Algorithm 3 is unconditionally stable for Equation (1) if and only if $\theta = 0$.*

**Proof.** Performing the substitutions $u_i^n \to \varepsilon_i^n$ described above, we obtain the following amplification factor for Algorithm 3:

$$G = \frac{1 - 2r\theta + 2r\cos(\gamma)}{1 + 2r(1-\theta)}.$$

In the $\theta = 0$ case, $G = \frac{1 + 2r\cos(\gamma)}{1+2r}$ is obviously between $-1$ and $1$ for any values of $\gamma$ and any nonnegative $r$, which implies unconditional stability. On the other hand, if $\gamma = \pi$, we have $G = \frac{1 - 2r(1+\theta)}{1 + 2r(1-\theta)}$, which is smaller than $-1$ for $r > \frac{1}{2\theta}$, and that implies instability for large time step sizes in the $\theta > 0$ case. $\square$

This statement means that unfortunately we cannot obtain a new unconditionally stable one-stage algorithm with the simple modification of setting the value of $\theta$ to a positive number.

For the investigation of stability of Algorithm 4, we substitute back conditions (15)–(18) into (10) to eliminate the $p$, $v_2$, $w_2$ parameters, set the external source term to zero and the reaction term to the homogeneous $K_i \equiv K$. With notations $\kappa = Kh$ and $\varphi = 1 - v_1$ this we obtain:

$$u_i^{n+1} = \frac{1}{\left[1 + \frac{r}{\lambda}\psi + \frac{\kappa}{2\lambda}\varphi\right](1 + r + w_1\kappa)} \left\{ \begin{array}{l} \frac{r^2}{2}\left(u_{i-2}^n + u_{i+2}^n\right) + r\left[1 + \frac{r}{\lambda}(\psi - \lambda) + \frac{\kappa}{2\lambda}[\varphi - 2\lambda(1-w_1)]\right]\left(u_{i-1}^n + u_{i+1}^n\right) + \\ \left(\begin{array}{l} 1 - r - \frac{r}{\lambda}[(r-2)\psi + \psi - r\lambda] - \frac{\kappa^2}{2\lambda}[\lambda(1-2w_1) - \varphi(1-w_1)] \\ +\kappa\left[(1 + \frac{r}{\lambda}\psi)(1-w_1) - \frac{1}{2\lambda}(\varphi + r(2\lambda(1-2w_1) - \varphi))\right] \end{array}\right)u_i^n \end{array} \right\}. \quad (20)$$

**Theorem 3.** *If $K = 0$, $\lambda + \theta_1 = 1$, and the conditions (15)–(18) hold, then Algorithm 3 is unconditionally stable for Equation (1).*

**Proof.** We take $K = 0$ in the previous expression of $u_i^{n+1}$, and introduce the notation $\eta = \frac{\psi}{\lambda} = \frac{1-\theta_1}{\lambda}$. With this we obtain

$$u_i^{n+1} = \frac{r^2}{2(1+r\eta)(1+r)}\left(u_{i-2}^n + u_{i+2}^n\right) + \frac{r(1-r+r\eta)}{(1+r\eta)(1+r)}\left(u_{i-1}^n + u_{i+1}^n\right) + \frac{1 - r - \eta r^2 + \eta r + r^2}{(1+r\eta)(1+r)}u_i^n. \quad (21)$$

This yields the following amplification factor:

$$G = \frac{1 + r(-1 + r + \eta - r\eta) + 2r(1 + r(-1 + \eta))\cos(\gamma) + r^2\cos(2\gamma)}{(1+r)(1+r\eta)}$$

Since $0 \leq \theta_1 \leq 1$, and $0 < \lambda \leq 1$, we have $\eta = \frac{1-\theta_1}{\lambda} \geq 0$. Now if $r \to \infty$, we have

$$G \to \frac{(1-\eta)[1 - 2\cos(\gamma)] + \cos(2\gamma)}{\eta},$$

which is guaranteed to be between $-1$ and $1$ if and only if $\eta = 1$, i.e., $\lambda + \theta_1 = 1$. Using this assumption, the $G$ function will have a simpler form:

$$G = \frac{1 + 2r\cos(\gamma) + r^2\cos(2\gamma)}{1 + 2r + r^2},$$

which is always in the interval $[-1, 1]$ due to the triangle inequality. $\square$

**Remark 1.** *From Equation (21) one can see that if the $\lambda + \theta_1 = 1$ holds, the value of $u_i^{n+1}$ does not depend on the parameters $\theta_1$ and $\lambda$. So, for the K = 0 case we will use $\lambda = 1$, which implies that the computer do not need to calculate linear combination (9), and the running times will be slightly shorter.*

Now we turn our attention to the case when the linear reaction term is nonzero. We will prove the stability of the method with the following parameters:

$$\lambda = \frac{1}{2}, \ \theta_1 = \frac{1}{2}, \ v_1 = 0, \ w_1 = 1, \tag{22}$$

which, if substituted to conditions (15)–(18), yields

$$p = 1, \ \theta_2 = \frac{1}{2}, \ v_2 = 1, \ w_2 = -1. \tag{23}$$

**Theorem 4.** *If conditions (22) and (23) hold, then Algorithm 3 is unconditionally stable for Equation (1) for arbitrary $K \geq 0$.*

**Proof.** Applying the assumptions of the theorem to (19) or (20) we obtain

$$u_i^{n+1} = \frac{1}{(1+r+\kappa)^2}\left\{ \frac{r^2}{2}\left(u_{i-2}^n + u_{i-2}^n\right) + r(1+\kappa)\left(u_{i-1}^n + u_{i+1}^n\right) + \left[1 + \kappa(1-2r) - \frac{\kappa^2}{2}\right]u_i^n \right\},$$

which yields

$$G = \frac{2 - \kappa(-2 + 4r + \kappa)}{2(1 + \kappa + r)^2} + \frac{2r(1+\kappa)}{(1+\kappa+r)^2}\cos(\gamma) + \frac{r^2}{(1+\kappa+r)^2}\cos(2\gamma).$$

The $G$ function is symmetric for $\gamma = \pi$, thus it is enough to examine the $0 \leq \gamma \leq \pi$ interval. First let us examine the limits of $G$. For $r \to 0$ we have

$$G \to \frac{2 + 2\kappa - \kappa^2}{2(1+\kappa)^2},$$

which function has a value of unity if $\kappa = 0$ and then it is monotonously decreasing with increasing $\kappa$ towards the limit value $-1/2$. For $r \to \infty$ we have $G \to \cos(2\gamma)$. For $\kappa \to \infty$ we have $G \to -\frac{1}{2}$. All of these values are finite and in the interval $[-1, 1]$.

Since $G$ is continuous, the limits are finite, the denominator cannot be zero, $G$ is bounded. Now it is enough to examine the condition $G \in [-1, 1]$ in the extremal values. We start with the variable $\gamma$ while keeping the other two variables arbitrary.

$$\frac{\partial G}{\partial \gamma} = -\frac{2r(1 + \kappa + 2r\cos(\gamma))}{(1+\kappa+r)^2}\sin(\gamma),$$

thus, extremal values can be present in three cases:

$$\gamma_1 = 0, \ \gamma_2 = \pi$$
$$1 + \kappa + 2r\cos(\gamma_3) = 0 \ \Rightarrow \ \cos(\gamma_3) = -\tfrac{1+\kappa}{2r} \, .$$

In the first case,

$$G\big|_{\gamma_1=0} = \frac{2(1+r)^2 + 2\kappa - \kappa^2}{2(1+\kappa+r)^2} = \frac{(1+r)^2 + \kappa - \kappa^2/2}{(1+r)^2 + \kappa + r\kappa + \kappa^2},$$

which is obviously in the interval $[-1, 1]$. In the second case,

$$G\big|_{\gamma_2=\pi} = \frac{(r-1)^2 + \kappa - \kappa^2/2 - 4\kappa r}{(1+\kappa+r)^2},$$

and it can be checked that the equation $G\big|_{\gamma_2=\pi} = 1$ and $G\big|_{\gamma_2=\pi} = -1$ have no solution for any positive values of $r$ and $\kappa$.

The third case gives an extremal value if $2r \geq 1 + \kappa$. We obtain the following function:

$$G_3 = G\big|_{\mathrm{Cos}\gamma_3 = -(1+\kappa)/2r} = \frac{1/2 - (\kappa+r)^2}{1 + 2(\kappa+r) + (\kappa+r)^2},$$

which can have values only in the interval $[-1, 1]$ again, which completes the proof. $\square$

**Remark 2.** *As the $|G|$ function is obviously continuous in its three variables with no singularities, if there is an extremal value above 1 at a point $(K^*, g^*, r^*)$, then there is a neighborhood of $(K^*, g^*, r^*)$ with values of $|G|$ also above 1. Any sufficiently dense set of points $(K_i, g_j, r_k)$ would intersect with this neighborhood. Thus, we independently verify the discussion above by examining $|G|$ numerically. We coded three embedded for loops, one to sweep through the values of each of the parameters K, g, and r in the allowed domain as follows*

$$K_i = 0.001 \times 1.01^i, \ i = 0 \ldots N_K, \ N_K = 20{,}000,$$
$$g_j = \tfrac{2j}{N_g} - 1, \ j = 0 \ldots N_g, \ N_g = 20{,}000,$$
$$r_k = 0.001 \times 1.01^k, \ k = 0 \ldots N_r, \ N_r = 20{,}000.$$

*During this process the value of $|G|$ has been calculated $8 \times 10^{12}$ times while K and r reached values larger than $10^{83}$, and it was found that $|G|$ does not exceed unity. Although this numerical procedure cannot be considered as an exact proof, we can conclude again that the algorithm is unconditionally stable if the conditions of Theorem 3 hold.*

**Remark 3.** *If we take $K \geq 0$ arbitrary, then we have proved the unconditional stability of Algorithm 3 only for $\lambda = \frac{1}{2}$. In fact, it would require enormous energy to thoroughly examine the full multidimensional parameter space. If the $\lambda = \frac{1}{2}$ assumption does not hold, the amplification factor can take values that are smaller than $-1$ in many cases. However, it does not mean that violating the assumptions always implies unstable behavior. In fact, according to a large number of numerical experiments, the value of G can be below $-1$ only if both Kh and r have a rather large value, typically larger than 20, and even in these cases G is still close to $-1$, for example $G = -1.002$, especially if $\lambda$ is not very close to 0. So, Algorithm 4 has very good stability properties in the practically relevant cases for $\lambda > 0.3$, as we will see in the next section.*

Although all the free parameters are fixed due to the analytical considerations, we will still consider $\lambda$ as a free parameter. It means that with conditions (22)–(23) we have Algorithm 5 with only one free parameter:

---

**Algorithm 5:** (Algorithm 4) For the diffusion-reaction-radiation Equation (1)

---

*Stage 1.* Take a partial time step $h_1 = \frac{h}{2\lambda}$ , $\lambda > 0$:

$$u_i^{\text{pred}} = \frac{\left(1 + r\left(1 - \frac{1}{\lambda}\right)\right)u_i^n + \frac{r}{2\lambda}\left(u_{i-1}^n + u_{i+1}^n\right) + q_i h_1}{1 + r + K_i h_1 + \sigma h_1 \left(u_i^n\right)^3}$$

*Stage 2.* Calculate the linear combination $u_i^{\text{pred}} = \lambda u_i^{\text{pred}} + (1 - \lambda)u_i^n$
Take a full time step:

$$u_i^{n+1} = \frac{(1-r)u_i^n + r\left(u_{i-1}^{\text{pred}} + u_{i+1}^{\text{pred}}\right) + q_i h + K_i h\left(u_i^{\text{pred}} - u_i^n\right)}{1 + r + K_i h + \sigma h\left(u_i^{\text{pred}}\right)^2 u_i^n}$$

---

### 2.3. Generalization for Arbitrary Grids

In the case when one has a general mesh and the material properties are functions of the space variables, the spatially discretized form of Equation (3) can be generalized as follows:

$$\frac{du_i}{dt} = \sum_{j \neq i} \frac{u_j - u_i}{R_{ij}C_i}. \tag{24}$$

here, $u_i$ refers to the cells of various shapes and properties with heat capacity $C_i$., while $R_{ij}$ is the thermal resistance between cells $i$ and $j$. If we use the notations $V_i$ for the volume of the cell, $A_{ij}$ and $d_{ij}$ for the surface between the cells and for the distance between the cell-centers, then these quantities can be calculated approximately as

$$C_i = c_i \rho_i V_i \text{ and } R_{ij} \approx \frac{d_{ij}}{k_{ij}A_{ij}}, \tag{25}$$

respectively. In our previous papers [9,11] this generalization procedure (which is based on, e.g., Chapter 5 of the book [40]) is explained in more details. Figure 1 can help the reader to visualize these quantities.

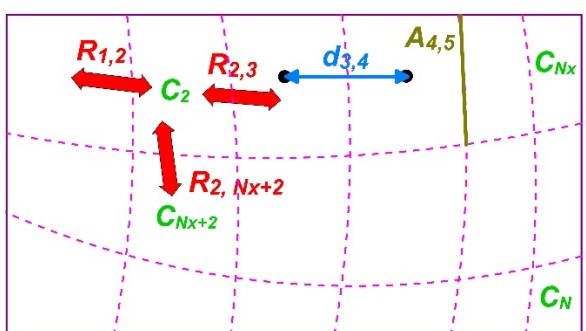

**Figure 1.** Arrangement of the generalized variables for the case when the mesh is not necessarily regular. The red double arrows are for conduction between cells with capacities $C_i$ and $C_j$ through the resistances $R_{ij}$.

In the general case of Equation (3), the nonzero elements of the matrix $M^D$ introduced in (13) can be given as:

$$M_{ij}^D = \frac{1}{R_{ij}C_i} \, , \, M_{ii}^D = -\sum_{j \neq i} M_{ij}^D.$$

We introduce the following notations

$$\tau_i = \frac{-1}{M_{ii}^D}, \, r_i = \frac{h}{2\tau_i} \text{ and } A_i = h \sum_{j \neq i} M_{ij}^D u_j^n = h \sum_{j \neq i} \frac{u_j^n}{C_i R_{ij}},$$

where $\tau_i \geq 0$ is the characteristic time or time constant of cell $i$, $r_i$ is the generalization of $r = \frac{\alpha h}{\Delta x^2} = -\frac{m_{ii}h}{2}$ (the usual mesh ratio in the case of the diffusion equation), and $A_i$ reflects the state of the neighbors of cell $i$. Now we can write the modified UPFD and our pseudo-implicit algorithms in the general case as follows (Algorithm 6):

---

**Algorithm 6:** (Algorithm 2) UPDF for the diffusion-reaction-radiation equation, general mesh-form

$$u_i^{n+1} = \frac{u_i^n + A_i + q_i h}{1 + 2r_i + K_i h + \sigma h (u_i^n)^3} \tag{26}$$

---

We emphasize that in Algorithm 7, $r_i = \frac{h}{2\tau_i}$ in both stages. We stress again that Algorithm 4 is proven to be unconditionally stable only for $\lambda = \frac{1}{2}$.

---

**Algorithm 7:** (Algorithm 4) 2-stage pseudo-implicit method for the diffusion-reaction Equation (1), general-mesh from

*Stage 1.* Take a partial time step $h_1 = \frac{h}{2\lambda}$, $\lambda > 0$, with the (26) formula:

$$u_i^{\text{pred}} = \frac{\left(1 + \left(1 - \frac{1}{\lambda}\right)r_i\right)u_i^n + A_i + q_i h_1}{1 + r_i + K_i h_1 + \sigma h_1 (u_i^n)^3}, \text{ where } A_i = h_1 \sum_{j \neq i} \frac{u_j^n}{C_i R_{ij}}.$$

*Stage 2.* We redefine $u_i^{\text{pred}}$ by calculating the linear combination

$$u_i^{\text{pred}} = \lambda u_i^{\text{pred}} + (1 - \lambda)u_i^n.$$

Take a full time step with the (26) formula:

$$u_i^{n+1} = \frac{(1-r_i)u_i^n + A_i + K_i h\left(u_i^{\text{pred}} - u_i^n\right) + q_i h}{1 + r_i + K_i h + \sigma h \left(u_i^{\text{pred}}\right)^2 u_i^n}, \text{ where } A_i = h \sum_{j \neq i} \frac{u_j^{\text{pred}}}{C_i R_{ij}}.$$

---

## 3. Numerical Results
### 3.1. Verification Using an Analitical Solution

We have constructed the following analytical solution of Equation (1) for $\alpha = 1$, $K = 2$ and $q(x,t) = \sigma t^4 e^{4x-4t} + e^{x-t}$:

$$u^{\text{exact}}(x,t) = te^{x-t}. \tag{27}$$

Here we reproduce this analytical solution numerically for $(t,x) \in [0.5, 1] \times [-1, 1]$ and $\sigma = 3$. The initial condition

$$u(x, t = 0.5) = 0.5e^{x-0.5},$$

and the Dirichlet boundary conditions at the ends of the interval

$$u(x = -1, t) = te^{-1-t}, \text{ and } u(x = 1, t) = te^{1-t}$$

are obtained using the analytical solution. The (global) numerical error is the absolute difference of the numerical solutions $u_j^{\text{num}}$ produced by the examined method and the reference solution $u_j^{\text{ref}}$ (which is the analytical solution here) at final time $t_{\text{fin}}$. We use these individual errors of the nodes or cells to calculate the maximum error:

$$\text{Error}(L_\infty) = \max_{1 \leq j \leq N} \left| u_j^{\text{ref}}(t_{\text{fin}}) - u_j^{\text{num}}(t_{\text{fin}}) \right|, \tag{28}$$

where the $L_\infty$ errors as a function of the time step size $h$ can be seen in Figure 2 for $\Delta x = 0.02$.

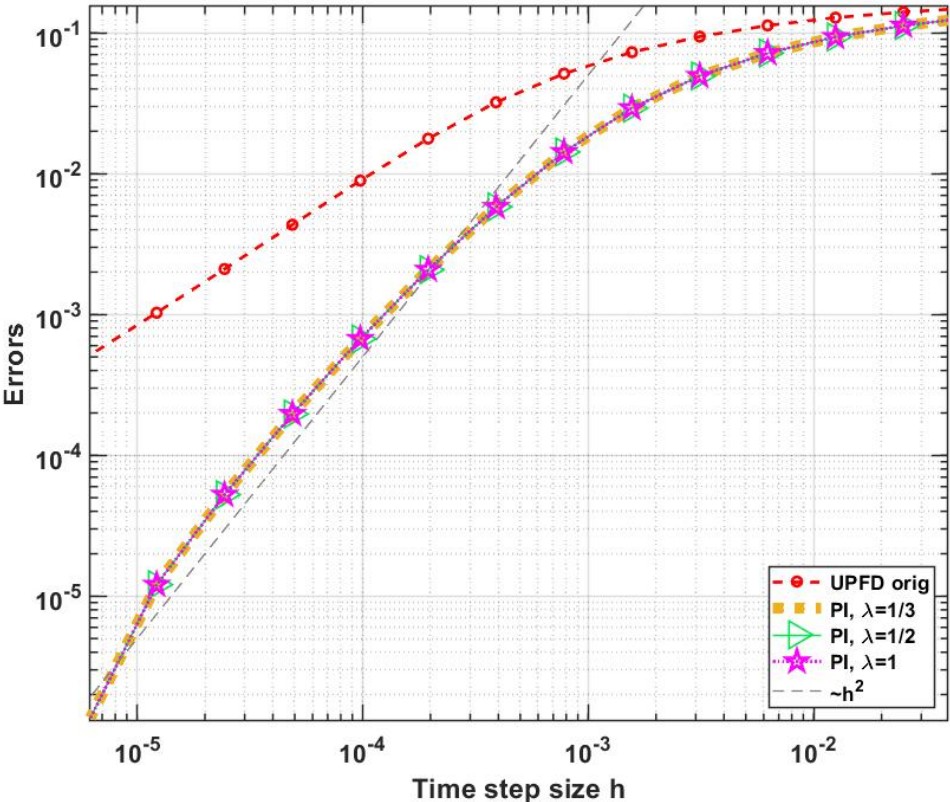

**Figure 2.** The $L_\infty$ errors as a function of time step size $h$ for the numerical solutions of Equation (1) in case of Algorithm 2 and the new pseudo-implicit Algorithm 4 for three different values of parameter $\lambda$. This parameter causes only very slight differences, thus the three curves almost coincide.

We have solved Equation (1) for several different space and time domains and values of the parameters $\sigma$ and $\Delta x$, and very similar curves have been produced for those cases as well. The new methods are completely stable, and the UPFD method is first order while the new pseudo-implicit methods are second order in the time step size. We note that in Section 3.4 this equation will be solved for a much larger radiation coefficient $\sigma$ as well, and it will be demonstrated that for a strongly nonlinear case and in a non-equidistant grid, the parameter $\lambda$ will be more relevant.

### 3.2. Comparison with Other Methods for a Large, Extremely Stiff System

In this subsection, we solve Equation (3) in a two space dimensional, topologically rectangle-structured mesh with $N = N_x \times N_z$ cells (see Figure 1 for visualization). The size of the system is fixed to $N_x = 100$ and $N_z = 120$, thus the total cell number is 12,000. Randomly generated cell capacities and thermal resistances

$$C_i = 10^{(\alpha_C - \beta_C \times rand)}, \ R_{x,i} = 10^{(\alpha_{Rx} - \beta_{Rx} \times rand)}, \ R_{z,i} = 10^{(\alpha_{Rz} - \beta_{Rz} \times rand)} \tag{29}$$

following a log-uniform distribution have been used, where the (pseudo)random number rand is generated by MATLAB for each quantity with a uniform distribution in the unit interval (0, 1). In this subsection, $K = 0$ and $\sigma = 0$, thus we deal with the linear heat equation and $M = M^D$. The exponents have been set to the following values:

$$\alpha_C = \alpha_{Rx} = \alpha_{Rz} = -3, \ \beta_C = \beta_{Rx} = \beta_{Rz} = 3,$$

which means that log-uniformly distributed values between 0.001 and 1000 have been given to the capacities and the resistances. Different random values have been generated for the initial conditions $u_i(0) = rand$ and the source term $q_i = 0.2 \times rand - 0.1$ as well. The final time of the simulation has been set to $t_{fin} = 0.2$.

We consider zero Neumann boundary conditions (isolated system). To implement this, we omit those terms of the sum in Equation (3) which have infinite resistivity in the denominator because of thermal isolation at the boundary. If the (nonzero) smallest and the largest absolute value eigenvalues of the system matrix $M$, defined in (11)–(13), are denoted by $\lambda_{\text{MIN}}$ and $\lambda_{\text{MAX}}$, then the stiffness ratio of the system can be given as $\lambda_{\text{MAX}}/\lambda_{\text{MIN}}$. On the other hand, $h_{\text{MAX}}^{\text{FTCS}} = \left| \frac{2}{\lambda_{\text{MAX}}} \right|$ exactly gives the maximum possible time step size for the FTCS (explicit Euler) scheme. This threshold is often called the CFL limit and it is valid for the second order explicit Runge–Kutta (RK) methods as well [41]. Above this time step size, the solutions will sooner or later explode due to instability. In the present case, the stiffness ratio is $2.3 \times 10^{11}$ and $h_{\text{MAX}}^{\text{FTCS}} = 1.03 \times 10^{-6}$, respectively. We will see that this implies serious under-performance of the conventional explicit methods, which are only conditionally stable.

In Sections 3.2 and 3.3, the reference solution is obtained using the ode15s built-in solver of MATLAB with sufficiently strict error tolerance $'\text{Tol}' = 10^{-12}$ (where $\text{Tol} \doteq '\text{AbsTol}' =' \text{RelTol}'$) and therefore high precision. Besides the $L_\infty$ error defined in (28), we also use the average error:

$$\text{Error}(L_1) = \frac{1}{N} \sum_{1 \leq j \leq N} \left| u_j^{\text{ref}}(t_{\text{fin}}) - u_j^{\text{num}}(t_{\text{fin}}) \right|,$$

and the so-called energy error:

$$\text{Error}(Energy) = \sum_{1 \leq j \leq N} C_j \left| u_j^{\text{ref}}(t_{\text{fin}}) - u_j^{\text{num}}(t_{\text{fin}}) \right|,$$

which, in case of heat transfer, gives the error in terms of energy.

The performance of the new algorithms was compared with the following methods coded by us. The original UPFD, the CNe [42], the 2-stage linear-neighbor (LNe) [11] and the CpC methods [31], and finally the well-known Heun method, also called as explicit trapezoidal rule, which may be the most common second order RK scheme [43]. Besides these, the well-established and professionally coded MATLAB solvers have been used for comparison purposes, namely:

- ode45, a fourth (fifth) order Runge–Kutta–Dormand–Prince method;
- ode23, second (third) order Runge–Kutta–Bogacki–Shampine method;
- ode113, 1 to 13 order variable-step and variable order VSVO Adams–Bashforth–Moulton solver;
- ode15s, a 1 to 5 order numerical differentiation formulas with VSVO, designed for stiff problems;
- ode23s, a second order modified Rosenbrock method;
- ode23t, uses the trapezoidal rule with a free interpolant;
- ode23tb, applies a trapezoidal rule in the first stage and a backward differentiation formula in the second one.

It is known that ode45, ode23 and ode113 uses explicit algorithms while the rest are implicit solvers. In case of the MATLAB solvers, the time step sizes cannot be determined directly, thus we set the tolerances instead, starting from an extremely large value, such as $\text{Tol} = 10^2$ until a small minimum value, usually $\text{Tol} = 10^{-6}$.

For the calculations where running times are measured, a desktop computer with an Intel Core i7-9700 CPU, 16.0 GB RAM is used, while the software is the MATLAB R2020b [44]. The total running time of the algorithms is measured by the built-in tic-toc function of that software.

We have examined the $L_\infty$, $L_1$ and energy errors as a function of the time step size $h$ and the running time. In Figure 3 we present the $L_1$ error as a function of $h$, while in Figure 4 one can see the $L_1$ errors vs. the total running times. Table 1 collects some results which have been obtained by the numerical schemes coded by us and the "ode" solvers of MATLAB. We set $\lambda = 1$ as it is explained in Remark 1.

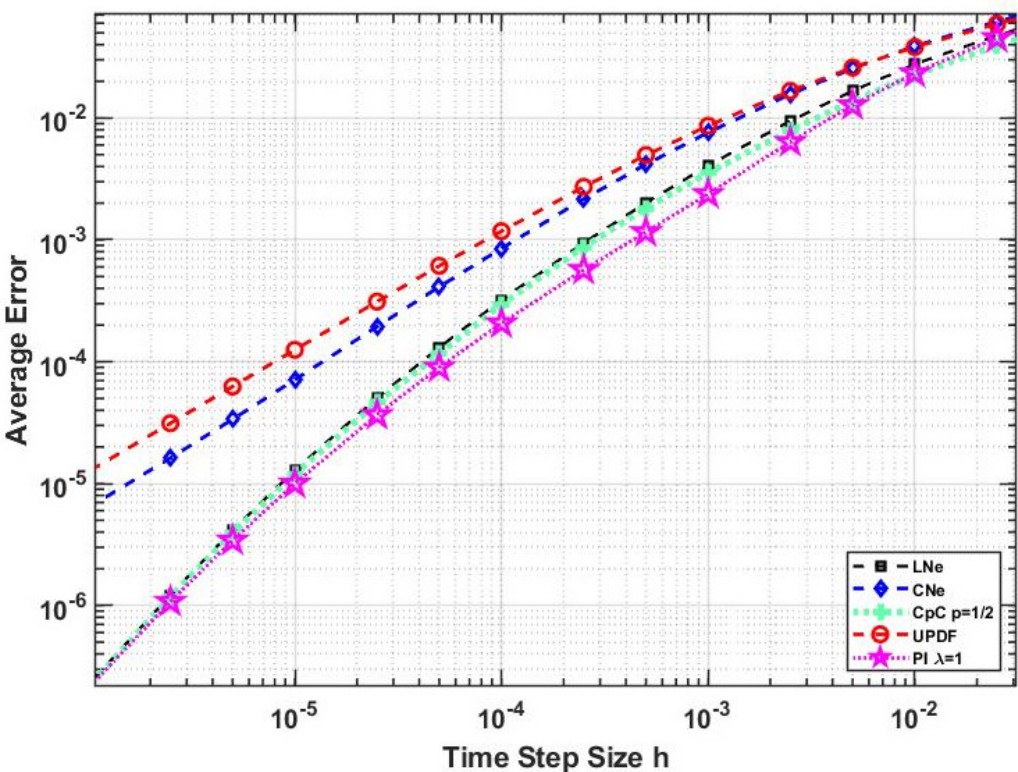

**Figure 3.** $L_1$ (average) errors as a function of the time step size of the new pseudo-implicit (PI) algorithm and some other methods for the first, extremely stiff system with $K = 0$, $\sigma = 0$.

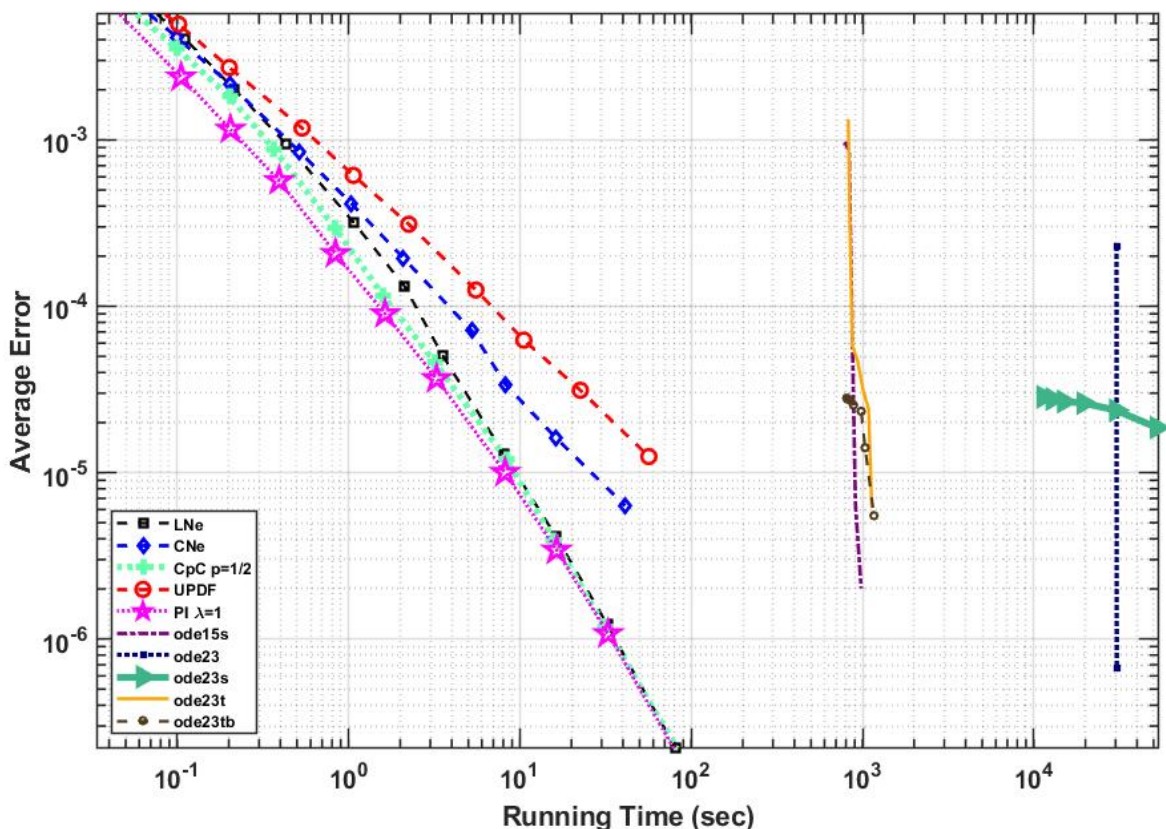

**Figure 4.** (average) errors as a function of the running time for the first (moderately stiff) system, in the case of the algorithms coded by us as well as of the MATLAB routines.

**Table 1.** Comparison of different algorithms for the extremely stiff system of twelve thousand cells.

| Numerical Method | Error($L_\infty$) | Error($L_1$) | Energy Error | Running Time (s) |
|---|---|---|---|---|
| ode23, Tol = $10^{-1}$ | $7.15 \times 10^{-3}$ | $6.68 \times 10^{-7}$ | $1.80 \times 10^{-5}$ | $3.04 \times 10^5$ |
| ode15s, Tol = $10^{-2}$ | $1.30 \times 10^{-3}$ | $7.33 \times 10^{-5}$ | $8.79 \times 10^0$ | $8.69 \times 10^2$ |
| ode23s, Tol = $10^{-2}$ | $4.33 \times 10^{-4}$ | $2.37 \times 10^{-5}$ | $2.80 \times 10^0$ | $3.02 \times 10^5$ |
| ode23t, Tol = $10^{-2}$ | $5.71 \times 10^{-4}$ | $3.14 \times 10^{-5}$ | $3.75 \times 10^0$ | $1.00 \times 10^3$ |
| ode23tb, Tol = $10^{-2}$ | $4.28 \times 10^{-4}$ | $2.33 \times 10^{-5}$ | $2.77 \times 10^0$ | $9.82 \times 10^2$ |
| UPFD, $h = 1 \times 10^{-6}$ | $2.20 \times 10^{-3}$ | $1.24 \times 10^{-5}$ | $4.86 \times 10^{-1}$ | $5.65 \times 10^1$ |
| Heun, $h = 1 \times 10^{-6}$ | $1.23 \times 10^{-11}$ | $3.79 \times 10^{-13}$ | $4.01 \times 10^{-8}$ | $1.05 \times 10^2$ |
| CNe, $h = 5 \times 10^{-6}$ | $5.85 \times 10^{-3}$ | $3.36 \times 10^{-5}$ | $1.28 \times 10^0$ | $8.28 \times 10^0$ |
| LNe, $h = 1 \times 10^{-5}$ | $2.70 \times 10^{-3}$ | $1.28 \times 10^{-5}$ | $3.58 \times 10^{-1}$ | $8.07 \times 10^0$ |
| CpC $p = 1/2$, $h = 2.5 \times 10^{-5}$ | $1.21 \times 10^{-2}$ | $4.61 \times 10^{-5}$ | $1.08 \times 10^0$ | $3.17 \times 10^0$ |
| PI$\lambda = 1$, $h = 2.5 \times 10^{-5}$ | $8.44 \times 10^{-3}$ | $3.66 \times 10^{-5}$ | $8.62 \times 10^{-1}$ | $3.26 \times 10^0$ |
| PI$\lambda = 1$, $h = 1 \times 10^{-5}$ | $2.54 \times 10^{-3}$ | $1.00 \times 10^{-5}$ | $2.41 \times 10^{-1}$ | $8.19 \times 10^0$ |
| PI$\lambda = 1$, $h = 5 \times 10^{-6}$ | $9.25 \times 10^{-4}$ | $3.41 \times 10^{-6}$ | $8.50 \times 10^{-2}$ | $1.63 \times 10^1$ |

One can see that the new scheme is slightly more accurate than the LNe and the CpC, and significantly more accurate than the first order UPFD and CNe methods. We note that the Heun method is not present in the figures, because it is convergent only below the CFL limit, which is lower than the time step sizes presented in the case of our methods. The explicit MATLAB solvers ode45 and ode113 were not able to provide any meaningful results and in the case of the ode23, it was a hard work to find those tolerances for which the method works, albeit very slowly. The implicit MATLAB routines performed usually much better, but even they are severely outperformed by the explicit and stable algorithms if running times are considered.

*3.3. Comparison with Other Methods for a Large System with Strong Nonlinearity*

In the second case study, we set $K_i = 3 \times rand$, $q_i = 2 \times rand$ and $\sigma = 1000$. The latter coefficient has been chosen so large because we would like to demonstrate the performance of the new method for a strongly nonlinear case, but the values of the variable $u$ are typically between zero and one, thus their fourth power is usually a rather small number. We give new values to the $\alpha$ and $\beta$ exponents:

$$\alpha_C = 3, \ \beta_C = 6, \ \alpha_{Rx} = \alpha_{Rz} = 3, \ \beta_{Rx} = \beta_{Rz} = 0.$$

We calculate the stiffness ratio and the CFL limit in two different ways, both of them without taking into account the nonlinear term. If we use the full $M$ matrix, we obtain that the stiffness ratio is $7.7 \times 10^5$, much smaller than in the previous case, while the CFL limit for the standard FTCS was $h_{MAX}^{EE} = 9.76 \times 10^{-4}$, which, we stress again, holds for the Heun method as well. If we use only the $M^D$ matrix instead of $M$, the stiffness ratio is $6.8 \times 10^9$, while the CFL limit is $h_{MAX}^{EE} = 9.75 \times 10^{-4}$. The reason behind these numbers is that the eigenvalues close to zero have been significantly increased (in absolute value) by the new reaction term while those with large absolute values remained almost the same.

All other parameters and circumstances, such as the size of the system and the range of the initial values are the same as in the previous subsection. We note that we were not able to adapt our previous methods CNe, LNe and CpC for the $K \neq 0$, $\sigma \neq 0$ case, nor when the advection term is present, without losing their order of convergence (that is why we started to develop the current methods), thus they are not presented in this and the next subsection. In Figures 5 and 6 the energy and the average errors are presented as a

function of the time step size and the total running time, respectively. In Table 2 we report the data that belong to this numerical experiment.

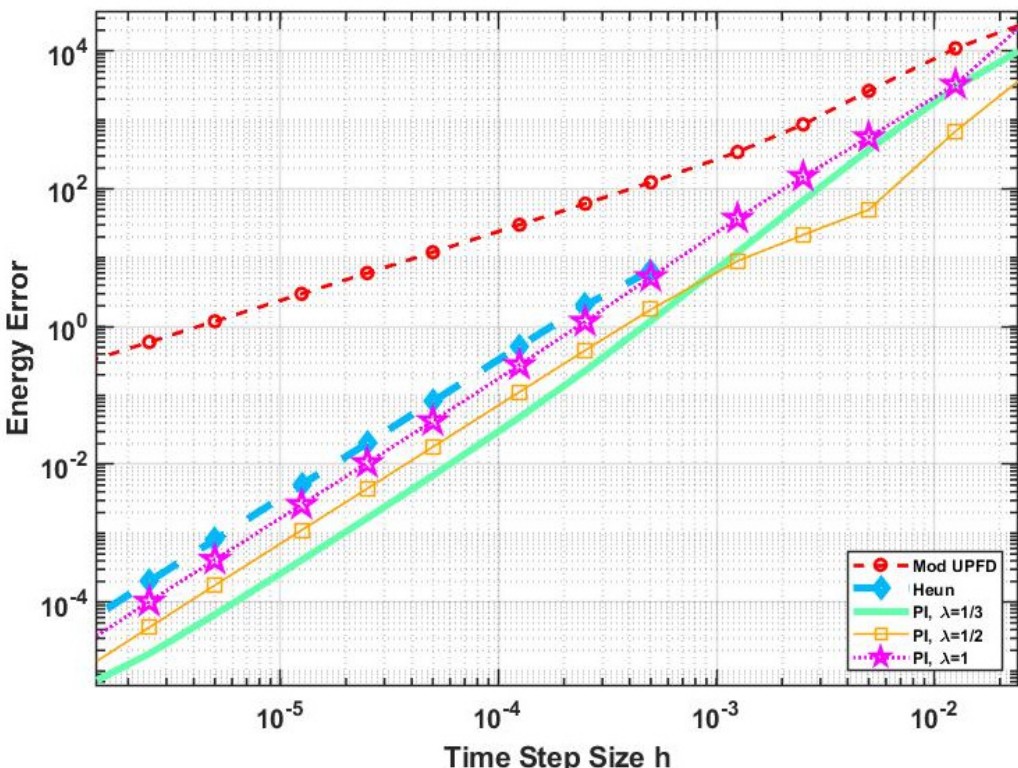

**Figure 5.** Energy errors as a function of the time step size for the second (very stiff) system, in the case of the UPFD Algorithm 2, the Heun method and the new PI algorithms.

**Table 2.** Comparison of different algorithms for the very stiff system of ten thousand cells.

| Numerical Method | Error($L_\infty$) | Error($L_1$) | Energy Error | Running Time (s) |
|---|---|---|---|---|
| ode45, Tol $= 10^{-1}$ | $4.49 \times 10^{-3}$ | $1.57 \times 10^{-6}$ | $8.54 \times 10^{-1}$ | $4.05 \times 10^1$ |
| ode23, Tol $= 10^{-1}$ | $7.97 \times 10^{-2}$ | $2.25 \times 10^{-6}$ | $1.17 \times 10^1$ | $1.67 \times 10^1$ |
| ode113, Tol $= 10^{-1}$ | $9.23 \times 10^{-2}$ | $1.07 \times 10^{-5}$ | $1.57 \times 10^0$ | $1.63 \times 10^1$ |
| ode15s, Tol $= 10^{-3}$ | $3.14 \times 10^{-4}$ | $4.94 \times 10^{-5}$ | $4.20 \times 10^1$ | $1.65 \times 10^3$ |
| ode23s, Tol $= 10^{-4}$ | $9.94 \times 10^{-5}$ | $2.31 \times 10^{-5}$ | $1.93 \times 10^1$ | $3.84 \times 10^4$ |
| ode23t, Tol $= 10^{-4}$ | $6.78 \times 10^{-5}$ | $1.78 \times 10^{-5}$ | $1.50 \times 10^1$ | $1.68 \times 10^3$ |
| ode23tb, Tol $= 10^{-4}$ | $1.41 \times 10^{-4}$ | $5.25 \times 10^{-5}$ | $4.48 \times 10^1$ | $1.67 \times 10^3$ |
| UPFD, $h = 5 \times 10^{-5}$ | $1.79 \times 10^{-4}$ | $1.22 \times 10^{-5}$ | $1.19 \times 10^1$ | $8.67 \times 10^{-1}$ |
| Heun, $h = 5 \times 10^{-4}$ | $1.12 \times 10^{-4}$ | $7.85 \times 10^{-6}$ | $6.50 \times 10^0$ | $3.33 \times 10^{-1}$ |
| PI $\lambda = 1/3$, $h = 1.25 \times 10^{-3}$ | $3.92 \times 10^{-4}$ | $1.53 \times 10^{-5}$ | $1.19 \times 10^1$ | $1.04 \times 10^{-1}$ |
| PI $\lambda = 1/2$, $h = 1.25 \times 10^{-3}$ | $3.85 \times 10^{-4}$ | $1.29 \times 10^{-5}$ | $8.97 \times 10^0$ | $1.06 \times 10^{-1}$ |
| PI $\lambda = 1/2$, $h = 5 \times 10^{-4}$ | $7.76 \times 10^{-5}$ | $2.58 \times 10^{-6}$ | $1.76 \times 10^0$ | $2.65 \times 10^{-1}$ |
| PI $\lambda = 1$, $h = 1.25 \times 10^{-3}$ | $4.17 \times 10^{-4}$ | $4.70 \times 10^{-5}$ | $3.71 \times 10^1$ | $1.00 \times 10^{-1}$ |

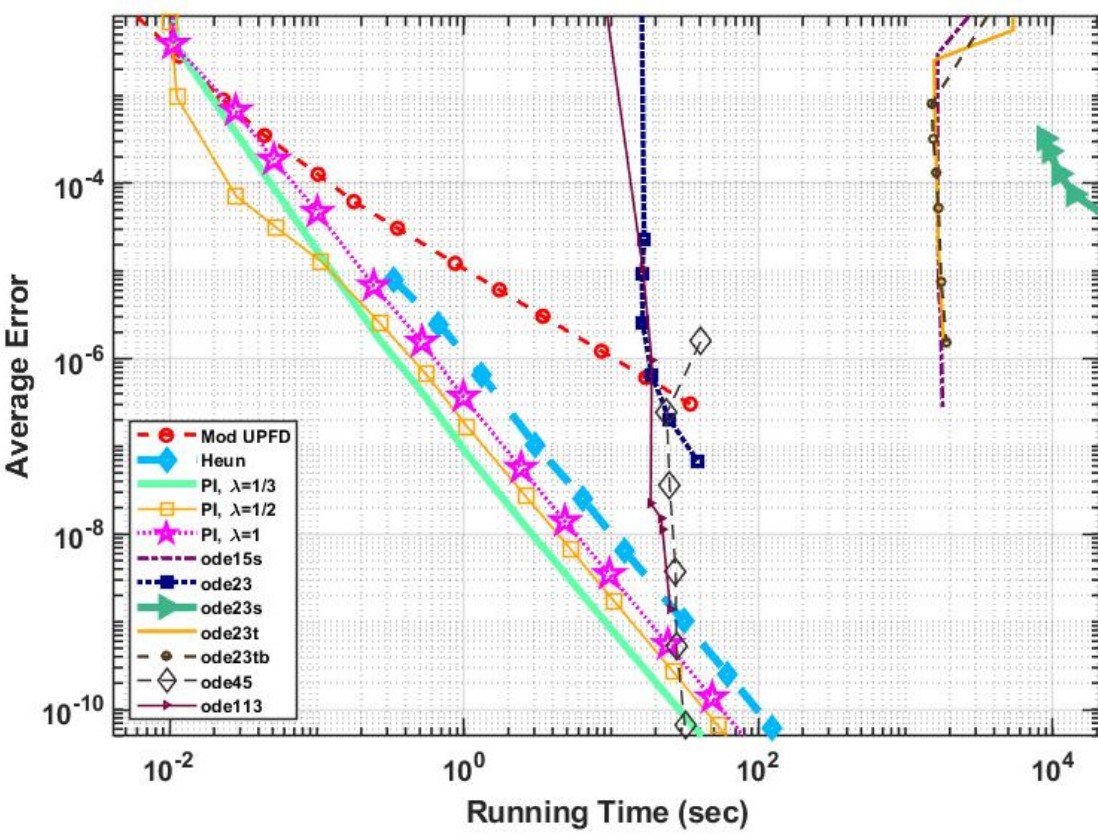

**Figure 6.** (average) errors as a function of the running time for the second (very stiff) system, in the case of the new algorithms and some other methods.

As it is expected, due to the larger CFL limit and weaker stiffness, the conventional explicit methods performed much better than the implicit ones, and especially the ode45 can compete with our methods if high accuracy is required. However, for low and medium accuracy requirements, the new pseudo-implicit method has the best performance.

### 3.4. Solution of the Advection-Diffusion-Reaction Equation

In the case of the advection–diffusion-reaction Equation (2), we found two meaningful possibilities to discretize the advection term. The first one applies the central difference formula for the first spatial derivative:

$$u_i^{n+1} = u_i^n + r\left[-2\theta u_i^n - 2(1-\theta)u_i^{n+1} + u_{i-1}^n + u_{i+1}^n\right] - \mu\frac{u_{i+1}^n - u_{i-1}^n}{2} - Khu_i^{n+1}.$$

with this we obtain

$$u_i^{n+1} = \frac{(1-2r\theta)u_i^n + r\left(u_{i-1}^n + u_{i+1}^n\right) - \frac{\mu}{2}\left(u_{i+1}^n - u_{i-1}^n\right)}{1 + 2r(1-\theta) + Kh}. \tag{30}$$

The second solution is what the original paper [30] proposes: the backward difference formula where the left neighbor is taken at the old, while the actual node is at the new time level:

$$u_i^{n+1} = u_i^n + r\left[-2\theta u_i^n - 2(1-\theta)u_i^{n+1} + u_{i-1}^n + u_{i+1}^n\right] - \mu\left(u_i^{n+1} - u_{i-1}^n\right) - Khu_i^{n+1},$$

which yields

$$u_i^{n+1} = \frac{(1-2r\theta)u_i^n + r\left(u_{i-1}^n + u_{i+1}^n\right) + \mu u_{i-1}^n}{1 + 2r(1-\theta) + \mu + Kh}. \tag{31}$$

Both treatments can be applied in the first and in the second stage, thus we have four combinations Algorithms 8–11 as listed below.

---

**Algorithm 8:** For the advection–diffusion-reaction equation

---

*Stage 1.* Take a partial time step $h_1 = ph$, $p > 0$ using Formula (30) and $\lambda = \frac{1}{2p}$:

$$u_i^{\text{pred}} = \frac{(1-2pr(1-\lambda))u_i^n + pr\left(u_{i-1}^n + u_{i+1}^n\right) - p\frac{\mu}{2}\left(u_{i+1}^n - u_{i-1}^n\right)}{1+r+Kh_1}$$

*Stage 2.* Calculate the linear combination $u_i^{\text{pred}} = \lambda u_i^{\text{pred}} + (1-\lambda)u_i^n$, and using this, take a full time step with the (30) formula:

$$u_i^{n+1} = \frac{(1-r)u_i^n + r\left(u_{i-1}^{\text{pred}} + u_{i+1}^{\text{pred}}\right) - Kh\left(u_i^n - u_i^{\text{pred}}\right) - \frac{\mu}{2}\left(u_{i+1}^n - u_{i-1}^n\right)}{1+r+Kh}$$

---

---

**Algorithm 9:** For the advection–diffusion-reaction equation

---

*Stage 1.* Take a partial time step $h_1 = ph$, $p > 0$ using Formula (31) and $\lambda = \frac{1}{2p}$:

$$u_i^{\text{pred}} = \frac{(1-2pr(1-\lambda))u_i^n + pr\left(u_{i-1}^n + u_{i+1}^n\right) + p\mu u_{i-1}^n}{1+r+p\mu+Kh_1}$$

*Stage 2.* Same as Stage 2 in Algorithm 8.

---

---

**Algorithm 10:** For the advection–diffusion-reaction equation

---

*Stage 1.* Same as Stage 1 in Algorithm 8.
*Stage 2.* Calculate the linear combination $u_i^{\text{pred}} = \lambda u_i^{\text{pred}} + (1-\lambda)u_i^n$, and using this, take a full time step with the (31) formula:

$$u_i^{n+1} = \frac{(1-r)u_i^n + r\left(u_{i-1}^{\text{pred}} + u_{i+1}^{\text{pred}}\right) - Kh\left(u_i^n - u_i^{\text{pred}}\right) + \mu u_{i-1}^n}{1+r+\mu+Kh}$$

---

---

**Algorithm 11:** For the advection–diffusion-reaction equation

---

*Stage 1.* Same as Stage 1 in Algorithm 9.
*Stage 2.* Same as Stage 2 in Algorithm 10.

---

We reproduce the following analytical solution of Equation (2) found in the paper of Appadu [32]:

$$u^{\text{exact}}(x,t) = e^{(\alpha+a-K)t-x}.$$

Here we examine the numerical solution for $(t,x) \in [0,\ 0.1] \times [0,\ 2]$ and $\alpha = 1$, $a = 2$, $K = 1$. The initial condition is $u^{\text{exact}}(x, t = 0)$, and we considered Dirichlet boundary conditions at the ends of the interval

$$u(x = -1,\ t) = te^{-1-t}, \text{ and } u(x = 1,\ t) = te^{1-t}.$$

The $L_\infty$ errors as a function of the time step size $h$ are presented in Figure 7 for $\Delta x = 0.02$ in the case of the original UPFD algorithm and the new Algorithms 5–8 above.

We found that the UPFD method is first order (as it is expected) while the new pseudo-implicit methods are second order in the time step size. However, the UPFD method and 2 of the new methods (those which use (31) in the second stage) have a problem with consistency. Moreover, there is an optimal time step size $h$ for the given space step size $\Delta x$ where some errors cancel each other and the algorithms are very accurate. In connection with the UPFD methods, earlier papers referred to these phenomena [30,32], and now we can see them in the case of the new pseudo-implicit methods as well.

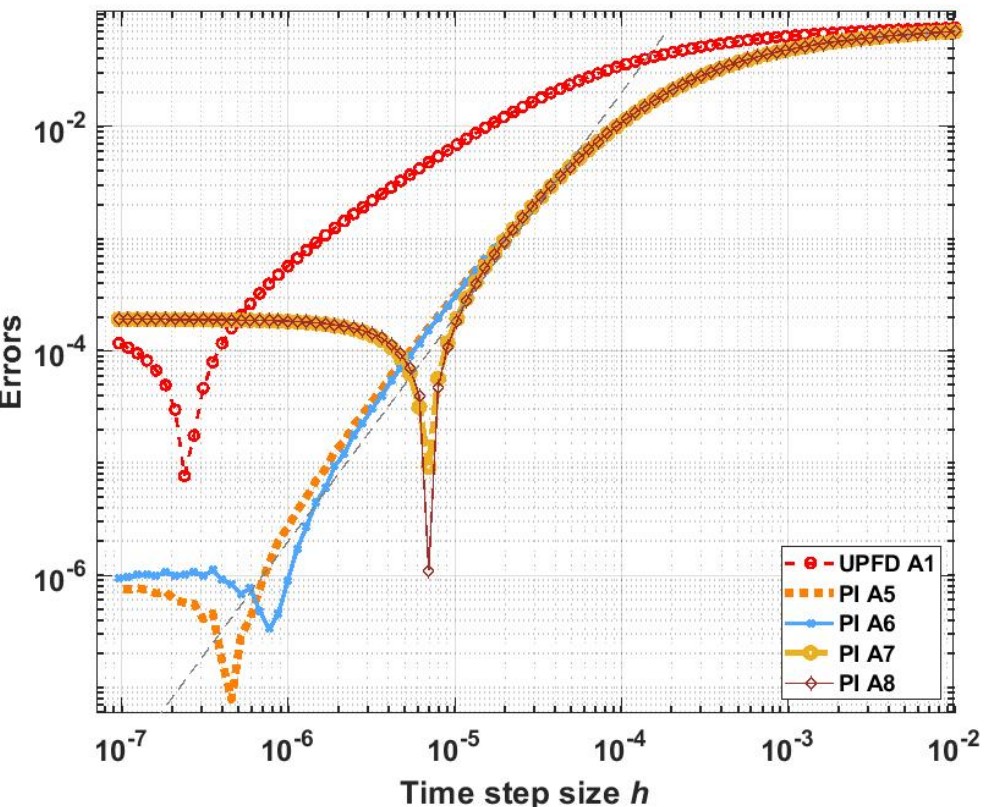

**Figure 7.** The $L_\infty$ errors as a function of time step size $h$ for the numerical solutions of Equation (1) in case of Algorithm 1 and the new pseudo-implicit Algorithms 8–11 for $\lambda = \frac{1}{2}$. The thin dashed grey line is proportional to $h^2$ again.

## 4. Discussion and Summary

In the current paper, we reached our goal to construct a fully explicit and stable numerical algorithm to solve the time-dependent diffusion (or heat) equation with linear and nonlinear reaction terms, where the latter represented heat loss due to radiation. Using the UPFD idea, we organized the theta-formula into a two-stage algorithm, where, in each stage, the latest available *u* values of the neighbors are used to make the originally implicit theta-formula completely explicit. We analytically proved for the linear case that the obtained method is second order in time step size and unconditionally stable.

For verification, an analytical solution of the nonlinear PDE was used. Then two 2-dimensional stiff systems containing 12,000 cells with discontinuous random parameters and initial conditions were constructed. The performance of the new algorithm as well as several other methods was examined for these systems. According to the numerical results, the new method is quite competitive. It is second order and stable for the nonlinear case as well, and it gives quite accurate results orders of magnitude faster than the professionally optimized MATLAB routines and it is more accurate than all other examined explicit and unconditionally stable methods. Although it is not positivity preserving as the original UPFD algorithm, it is stable for relatively large time step sizes as well, even if the nonlinearity is strong. Moreover, it is easy to implement and can be applied for unstructured grids as well. The conclusion is that this new pseudo-implicit algorithm has the most important advantages of the conventional explicit and the implicit methods at the same time.

In the near future, we are going to search for higher order and thus even more accurate versions of these algorithms and adapt these to nonlinear parabolic equations as well. We also plan to systematically investigate the application of these methods to the advection–diffusion-reaction Equation (2), and, also, to similar nonlinear equations like the Burgers–Fisher and the Burgers–Huxley equations. Moreover, we have started to consider applying these methods to real-life engineering problems, most importantly heat transfer

by convection, conduction, and radiation in buildings [45] and solar panels [46], to increase energy-efficiency and therefore to contribute to the prevention of the climate-change.

**Author Contributions:** Conceptualization, methodology, supervision and resources, E.K.; software, Á.N.; validation, E.K. and H.K.J.; formal analysis (proofs), J.M.; investigation, H.K.J. and A.H.A.; data curation, Á.N.; writing—original draft preparation, E.K. and J.M.; writing—review and editing, Á.N.; visualiza-tion, A.H.A. and H.K.J.; project administration, E.K. and Á.N. All authors have read and agreed to the published version of the manuscript.

**Funding:** This research was supported by the EU and the Hungarian State, co-financed by the ERDF in the framework of the GINOP-2.3.4-15-2016-00004 project.

**Institutional Review Board Statement:** Not applicable.

**Data Availability Statement:** Data is available at the following link https://github.com/Drendre/Pseudo-Implicit-method-codes-data, accessed on 25 November 2021.

**Conflicts of Interest:** The authors declare no conflict of interest.

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
