# Peer review of "Explicit Stable Finite Difference Methods for Diffusion-Reaction Type Equations"

_mathematics, doi:10.3390/math9243308_

Round 1
Author Response
Dear reviewer,
Thank you for the constructive comments and suggestions. We have tried to follow all of them as explained below.
We have now resolved the abbreviation UPFD in the abstract, before the keywords.
We give more information (ISBN and doi numbers) and corrected the required letter-cases in the list of references (and we hope that during the proofreading process, any remaining formal mistakes will be fixed). Moreover, we included some additional citations, one of them for the used MATLAB version.
We simplified the appearance of some long formulas by introducing some abbreviations.
Finally, but most importantly, we added a few words into Remarks 2 and 3 to indicate that these are not strict statements.
Reviewer 2 Report
In this manuscript, the authors proposed an explicit stable finite difference methods for diffusion-reaction type equations. By the iteration of the theta-formula and treating the neighbors explicitly authors construct a new 2-stage explicit algorithm to solve partial differential equations containing a diffusion term and two reaction terms. It is of high importance that authors verified the proposed method by reproducing an analytical solution with high accuracy.
This research is of great importance in the field of numerical solving the partial differential equations (PDE), particularly those of parabolic type.
Overall, the whole article is a good-written one with consecutiveness, strict logic, affluent datum, and clear consecution. I only suggest authors to add in the introduction section some more references which deal with numerical solutions of similar types of PDEs, for example:
Sadia Akter Lima, Md. Kamrujjaman, Md. Shafiqul Islam, Numerical solution of convection–diffusion–reaction equations by a finite element method with error correlation, AIP Advances 11, 2021, 085225.
- Urošević, D. Nikezić, Radon transport through concrete and determination of its diffusion coefficient. Rad. Prot. Dosimetry 104, No. 1, 2003, pp. 65-70.
- M. Ivanović, M. Svičević, Savović, Numerical solution of Stefan problem with variable space grid method based on mixed finite element/finite difference approach, International Journal of Numerical Methods for Heat and Fluid Flow, Vol. 27, No. 12, 2017, pp. 2682-2695.
- Savović, A. Djordjevich, Numerical solution of diffusion equation describing the flow of radon through concrete, Applied Radiation and Isotopes, Vol. 66, No. 4, 2008, pp. 552-555.
Author Response
Dear reviewer, thank you for your positive opinion about our work.
We have found all the papers you kindly suggested interesting and have included in the references, as well as other recent papers which deal with the numerical solutions of similar types of PDEs.
Reviewer 3 Report
Please see the attached file.

Author Response
Dear reviewer,
Thank you for spending your precious time to help us to improve our manuscript.
We tried to follow all your recommendations.
We looked through the manuscript again to improve its grammar, including the problems you kindly pointed out. There is one exception about the citation style, where we have followed the template of the journal: "For embedded citations in the text with pagination, use both parentheses and brackets to indicate the reference number and page numbers; for example [5] (p. 10), or [6] (pp. 101–105)." If any other formal problems or typos remained in the manuscript, they will be corrected during the proofreading process.
We provide a point-by-point response to your other comments as follows:
13. We have given some more details about why our formula preserves positivity.
15. We included the following words: "where M is defined in (11)-(13)," to clarify the meaning of M.
16.-18. We added two references on the von Neumann stability analysis and some extra words in different places to help the readers to understand more easily, which is our interest, obviously.
19. The figures are produced in a standard way using MATLAB and saving in jpg. We know that they are not perfect and continuously search for better alternatives, but currently, we have no better ideas and according to the editor, our figures meet the requirements of the journal.
20. We have added some extra papers into the literature, including what you and other reviewers proposed, as well as some other very recent papers, indexed by the main databases.
Thank you again for your advice and hope that you'll find the new version of the manuscript satisfactory.
Kind regards,
The authors.
Round 2
Reviewer 3 Report
Accept